# Exploring Compassion towards Laboratory Animals in UK- and China-Based Undergraduate Biomedical Sciences Students

**DOI:** 10.3390/ani13223584

**Published:** 2023-11-20

**Authors:** Richard Fitzpatrick, Nicola Romanò, John Menzies

**Affiliations:** 1School of Biological Sciences, University of Edinburgh, Edinburgh EH9 3FD, UK; rfitzpa2@exseed.ed.ac.uk; 2Centre for Discovery Brain Sciences, Edinburgh Medical School: Biomedical Sciences, University of Edinburgh, Edinburgh EH8 9XD, UK; nicola.romano@ed.ac.uk; 3Zhejiang University-University of Edinburgh Joint Institute, Zhejiang University School of Medicine, Zhejiang University, Haining 310058, China

**Keywords:** laboratory animals, compassion, biomedical sciences, students, culture of care

## Abstract

**Simple Summary:**

Research and teaching in biomedical sciences has led to huge advances in human wellbeing. However, some research activities require the use of non-human animals and can entail animal suffering. This suffering is obviously detrimental to the animal but can also negatively impact learners’ and researchers’ emotions or weaken the validity of the activity’s scientific outcomes. Taking a more compassionate approach in these research activities has the potential to address these issues, but compassion in the human–animal relationship is often only indirectly addressed in biomedical sciences teaching, and we need new ways of exploring this. Here, we developed a survey to measure compassion towards laboratory animals and used it in groups of biomedical sciences students in the UK and China. In line with other studies on compassion, we found that females are more compassionate than males and that nationality/culture-related differences are present. We found that students with lower levels of compassion are more likely to accept animal suffering in a research setting, and while exploring whether specific beliefs about animals’ mental capacities were linked to compassion, we found that a belief that animals are conscious is associated with higher levels of compassion. This survey can be used to investigate compassion towards lab animals, including investigations of ways to enhance compassion, and may help us create a more compassionate and scientifically robust setting for teaching, learning and research activities involving animals.

**Abstract:**

Taking a compassionate approach to the non-human animals used in biomedical research is in line with emerging ideas around a “culture of care”. It is important to expose biomedical sciences students to the concept of a culture of care at an early stage and give them opportunities to explore related practices and ideas. However, there is no simple tool to explore biomedical sciences students’ attitudes towards laboratory animals. Accordingly, there is little understanding of students’ feelings towards these animals, or a means of quantifying potential changes to these feelings. We developed a 12-item questionnaire designed to explore compassion (the Laboratory Animal Compassion Scale; LACS) and used it with UK-based and China-based samples of undergraduate biomedical sciences students. In the same samples, we also explored a harm–benefit analysis task and students’ beliefs regarding some mental characteristics of laboratory animals, then drew correlations with the quantitative measure of compassion. Compassion levels were stable across years of study and were not related to students’ level of experience of working with laboratory animals. We observed a higher level of compassion in females versus males overall, and a higher level overall in the UK-based versus China-based sample. In a task pitting animal suffering against human wellbeing, students’ compassion levels correlated negatively with their acceptance of animal suffering. Compassion levels correlated positively with a belief in animals being conscious and possessing emotions. These data are in line with studies that show compassion is gender- and nationality/culture-dependent, and points to links between compassion, beliefs, and choices.

## 1. Introduction

Research and teaching in biomedical sciences leads to advances in human health. However, this requires the use of tens of millions of non-human animals in experimental work, and much of this work entails animal suffering [1] By “suffering”, we do not refer only to the impact of regulated scientific procedures; this term can also encompass a capability approach to laboratory animals’ experiences in their day-to-day lives, including limitations on the expression of natural behaviours [2]. This suffering, at least in commonly used species like mice and rats, seems to be accepted by the public [3] and by the scientists carrying out such work [4], likely because of a belief that animal suffering is outweighed by the future health benefits to humans. In other words, a potential reduction in human suffering is apparently readily prioritised over animal suffering [5]. Ideas that animals are not (or at least less) sentient, are less susceptible to suffering, or have fewer or weaker cognitive or social capabilities compared to humans may underpin this acceptance of animal suffering [5,6,7,8], particularly if the animals involved are seen as evolutionarily distant from humans [9] or—like rats and mice—perceived in the non-scientific context as pests [10,11]. Such beliefs provide fertile ground for a non-compassionate approach to laboratory animal use where the generation of knowledge may be prioritised over animal suffering [12]. However, this suffering is obviously detrimental to the animal experiencing it, and can also impact the wellbeing of learners, researchers, and the animals’ carers [13], as well as the validity of the scientific outcomes [14,15].

Biomedical sciences students are rarely given opportunities to discuss taking a compassionate approach to animal research [16]. Teaching activities normally focus on the legal and regulatory frameworks that govern animal use at national and institutional levels. The teaching of ethical aspects in biomedicine is sometimes limited to a deceptively simple consequentialist/utilitarian view where animal research is taken for granted as morally correct in most circumstances, and that animal suffering is something a student should accept as normal and justified and be obliged to engage with. These anthropocentric utilitarian arguments are often delivered without a great deal of scrutiny, and students are rarely given the opportunity to challenge these ideas and explore alternatives, for example, a virtue ethics or “ethics of care” approach, where an individual’s compassion, generosity, kindness, or respect are of a stronger influence in comparison to a person’s right to carry out a certain action and/or the ethical nature of the outcomes of that action [17,18,19]. In addition, students are often implicitly taught they must be detached and professional when working with animals, and students who demonstrate kindness and care may be perceived negatively as being sentimental or unprofessional [20].

However, emerging ideas around a “culture of care” are beginning to challenge these conceptions. A culture of care is defined as a commitment to improving the transparency, welfare, scientific quality, and care of people who care for and work with animals [21]. This manifests in many ways, including improvements to the animal’s environment [22], their interactions with people [23], positive animal welfare approaches [24], and a deeper consideration of human–animal relationships [25,26]. Part of this deeper consideration is an exploration of how people feel about animals. Compassion has been defined as tolerating one’s own uncomfortable feelings when recognising and understanding suffering, and a motivation to alleviate suffering [27]. Efforts to quantify compassion towards animals have mainly used Likert scale responses to specific statements. To the best of our knowledge, compassion towards laboratory animals has not been studied in biomedical sciences students, but data on veterinary students suggest that compassion towards the animal they work with is relatively high in early years, but declines, particularly in male students, as they progress through their programme [28,29,30,31]. A variety of explanations were put forward for this, including a “hardening” of attitude, “counter-anthropomorphising animals, possibly as a way of coping with the moral conflict and emotional distress” [32], and/or compassion fatigue.

To address this gap in understanding, we developed and validated a survey to explore compassion towards laboratory animals. We deployed the survey with undergraduate students on biomedical sciences degree programmes based in either the UK or China. In the same cohorts, we also explored whether compassion scores were associated with choices made in a simple harm–benefit analysis task, and to explore potential links between beliefs and attitudes, and whether compassion scores were associated with beliefs about consciousness and emotion in different laboratory animals.

## 2. Materials and Methods

We developed a 12-item scale to assess compassion towards laboratory animals based loosely on Elizabeth Paul’s Animal Empathy Scale [33]. Each item in our Laboratory Animal Compassion Scale (LACS; see Appendix A) comprised a statement, with six “pro-compassionate” and six “anti-compassionate” statements in total. A Likert scale was used to capture responses and converted to a numerical value. These values were summed for each participant to generate their compassion score.

Next, we used a simple harm–benefit analysis of human/animal suffering with the aim of correlating responses with participants’ compassion scores. We asked participants to imagine designing biomedical research programmes that would provide a simple, effective, and safe treatment that will prevent the onset of three different human diseases: psoriasis, type 2 diabetes (T2D), and fibrodysplasia ossificans progressiva (FOP; [34]). We chose these diseases to represent a common disease with a low impact on mortality, a common disease with a more serious impact on patients, and a rare but extremely serious disease, respectively. Each imagined study would use the same and scientifically appropriate number of rats. We asked participants what level of animal suffering was acceptable in order to achieve the scientific outcomes of each study. We used the UK Home Office classifications of suffering as answer options: no deliberate suffering, mild suffering, moderate suffering, or severe suffering.

The factors that contribute to non-human animals’ moral standing are unclear [5], but we speculated that believing a species is conscious and experiences emotions—in other words, the animal can subjectively experience suffering and there is an emotional valency to this experience—may be associated with a higher degree of compassion towards that animal. To explore whether such beliefs were associated with compassion scores, we asked participants about whether they believed that they themselves, other humans, and two commonly used laboratory animals—rats and fruit flies—were conscious and/or possessed emotions. We gave simple definitions of phenomenal consciousness [35] and emotion in the survey. We asked about these mental capabilities in a simple and generalised way because we did not want participants to be distracted by the consideration of complex capabilities associated with human consciousness and emotion; for example, whether self-awareness is required in order for a species to be conscious, or whether another species’ subjective experience of suffering would need to be similar in nature to a human experience of suffering in order for it to be worthy of consideration.

We defined consciousness as the ability to have subjective experiences. We gave some examples of such experiences, and clarified that self-awareness was not included in our definition. We asked about their belief in human consciousness to benchmark the understanding of the concept in responders, expecting that almost all responders would state that they themselves were conscious. The survey defined emotion as a conscious and subjective mental state that is sometimes accompanied by physiological or behavioural changes, and stated that emotions can be positive or negative, strong or weak, long-lasting or short-lasting, directed to something in particular or undirected, giving simple examples for each. The responses were captured using a Likert scale.

Lastly, we asked responders to report their gender, whether they were based at the University of Edinburgh (UK) or at the University of Edinburgh-Zhejiang University Joint Institute (ZJE; Haining, China). Both cohorts were matriculated on four-year B.Sc. Hons degree programmes in subjects in the biomedical sciences field. The UK-based students were taught by UK-based staff and the ZJE students were taught by UK- and China-based staff, with a small number of UK-based staff teaching at both sites. All students were taught in English. Both programmes featured a similar amount and level of taught material on the use of animals in science. We did not ask the responders to self-report their ethnicity or heritage in the survey, but the UK-based student population mainly comprised people with a “Western” background, whereas the China-based student population was >99% Chinese nationals. We also asked the responders to report their year of study, and their self-reported level of experience of working with laboratory animals.

The survey responses were recoded as integers. For the statements about compassion toward laboratory animals, responses to pro-compassionate statements were scored on a scale from 0 (“Strongly disagree”) to 5 (“Strongly agree); anti-compassionate statements were reverse-coded, and a compassion score (range 0–60) was generated by summing the scores for each participant. Reponses in the harm–benefit analysis were scored from 0 (no deliberate suffering) to 3 (severe suffering). The diseases were coded as ordered factors using two scores ranging from 1 to 3 for the number of humans affected and the severity of the disease. Questions about consciousness in animals were scored from –3 (I am certain that … not …) to 3 (I am certain that …). We used a generalised linear model (GLM, using the R *glm* function) to analyse the compassion data, and ordinal regression (using the R *ordinal::clmm2* function) for the harm–benefit analysis and the consciousness/emotion responses. All models included the site (UK or China), gender, year of study, and experience working with animals (plus their interaction, as appropriate) as factors.

The validity of the responses to the compassion statement was visually evaluated using a principal component analysis (PCA; calculated using *prcomp*). All analyses were performed using R 4.2.3.

## 3. Results

### 3.1. Participant Demographics

All undergraduate students matriculated on biomedical sciences programmes at the University of Edinburgh or at ZJE were eligible to take part. The total number of eligible students was 1485, comprising 1056 UK-based students and 429 China-based students. Data were collected using an online survey between December 2021 and February 2022. The survey was advertised during classes and in online programme-wide announcements.

All participants gave informed consent. There was a total of 118 participants: 55 China-based, and 63 UK-based, representing 13% and 6% of the respective populations. The responders’ gender balance reflected the population at both sites: 53% female responders from the 55% female China-based population, and 71% female responders from the 77% female UK-based population, indicating little gender-based self-selection bias. Three responders (2.5%) reported their gender as “other”, and seven responders (6%) selected “prefer not to say”. These responders were included in the overall GLM analysis, but because of the small proportion of these categories, we report gender-related data only for responders that selected either male or female. Most responders from both sites were Year 1 students (52% overall; Figure 1), with a roughly equal representation for Years 2–4 in the China-based sample but diminishing representation from Year 2 to Year 4 in the UK-based sample. As expected, most responders self-reported low levels of experience of working with laboratory animals (Figure 1), though experience levels were more broadly distributed in the China-based responders. The level of experience correlated strongly with the year of study (Appendix A, Pearson’s R = 0.530, *p* = 7.9 × 10^−10^).

### 3.2. Survey Validity

A principal component analysis was used to evaluate the validity of the compassion statements (see Appendix A for the statements, and Appendix A). When the first two principal components were plotted, the pro-compassionate statements were clearly separated from the anti-compassionate statements, indicating validity. One possible exception was the pro-compassionate statement “Laboratory rodents should be allowed to live in an environment that allows them to express natural behaviours. For example, social interactions, building nests and digging” (Appendix A).

### 3.3. Effect of Year of Study, Experience, Gender, and Site

A GLM analysis showed that the compassion scores were not associated with the year of study or self-reported experience at either site (Figure 2). However, the overall compassion scores were different by gender (Figure 2; Table 1), with higher compassion scores in females. The overall compassion scores were different by site (Figure 2; Table 1), with the UK-based responders showing a significantly higher score than the China-based responders. No ceiling or floor effects were observed.

### 3.4. Harm–Benefit Analysis Task

The proportion of responders who accepted severe animal suffering increased as the implicit severity of the disease increased (Figure 3; Table 2), with only 6% of responders agreeing that severe suffering would be justified in research that prevented psoriasis, but 21% of responders agreeing that severe suffering would be justified in research that prevented FOP. There was no relationship between the year of study or experience with the mean acceptable level of suffering (Table 2), but the mean acceptable level was consistently higher in the China-based responders when expressed by either year of study, gender, or experience, except in responders with the highest self-reported level of experience, where the mean level was higher in the UK-based students (Figure 3). This was confirmed though the GLM analysis, which showed that the only significant factors were the site and disease severity (Table 2). There was a significant negative correlation between the compassion scores and acceptable levels of suffering for all three diseases (Figure 4, r_Psoriasis_ = −0.454, r_Diabetes_ = −0.373, r_FOP_ = −0.445, *p* < 10^−5^ for all diseases).

### 3.5. Beliefs about Animal Consciousness and Animals’ Emotional Capabilities

A total of 86% of the responders were certain or fairly sure that they themselves are conscious. This level of certainty was reduced when considering consciousness in other humans (73% were certain or fairly sure), reduced further by roughly the same magnitude for rats (61%), and reduced further for fruit flies (32%; Figure 5). A qualitatively similar pattern was seen for beliefs about species’ possession of emotion (Figure 6). The GLM analysis showed an effect of the year of study: later-years students were more likely to ascribe consciousness and emotion to all species, though the China-based students were less likely to attribute emotions to all species compared to the UK-based students (Table 3). There was a positive correlation between the compassion scores, a belief in emotions in rats and fruit flies, and a belief in consciousness for rats (Figure 7; Appendix A).

## 4. Discussion

To our knowledge, this is the first study to explore compassion towards laboratory animals in biomedical sciences students. We surveyed students across all years of study at two comparable sites: one in the UK and one in China. A principal component analysis indicated that the survey is valid. We have not yet carried out a test–retest to assess consistency.

In contrast to studies on compassion towards animals in veterinary students, we did not see a reduction in compassion scores in later years of study. Instead, the compassion scores were stable across all four years of study. Importantly, undergraduate students—particularly Year 1 students who make up the majority of our sample—have relatively little direct exposure to animal experiments. Accordingly, they may have a relatively limited understanding of the harms and benefits of animal use, hence the inclusion of a question on the students’ levels of experience with lab animals. However, increasing levels of experience were not associated with a change in compassion scores. The number of responders with higher levels of experience was low (only 9% overall) and may be insufficient to detect a change in compassion after exposure to animal studies. It may be of interest to use this survey in a longitudinal study with biomedical sciences postgraduate students, comparing compassion scores over time in those working closely with animals versus those who do not. In this way, we could ask whether direct and sustained exposure to laboratory animals has an effect on compassion over time and explore which factors might contribute to any observed changes.

We saw lower compassion scores in male versus female students. This is not surprising given that other studies have observed higher levels of compassion or empathy towards animals in female responders [36,37,38]. The overall compassion scores in the China-based cohort were lower than those in the UK-based cohort, and the average allowable level of suffering in the harm–benefit analysis was higher in the China-based cohort compared to the UK-based cohort. These site-dependent differences may have a number of explanations. Compassion may be defined or expressed in different ways in different cultures, and/or moral standing may be extended to animals in a culturally dependent way. In other words, it is unlikely that all cultures and societies value the wellbeing of non-human species in the same ways. Accordingly, cultural differences in attitudes towards animals have been reported. Su et al. reported that while 57% of responders from a European country supported the use of rats and mice in research, a higher number (86%) of responders from China supported this use [39]. In a study of students’ cultural attitudes towards animals in 11 countries [40], Chinese responders had the lowest score in a measure of concern for animal welfare, while UK responders had the second-highest score. However, the levels of concern about the use of animals in scientific research were similar between UK and Chinese responders [38], and the levels of compassion fatigue in laboratory animal carers are approximately equal in the European Union and China [41]. Interestingly, other studies show that Chinese responders attribute a greater degree of sentience to rats (but not, for example, to pigs or octopus) compared to responders from the UK [38,40]. 

Importantly, a group displaying a lower compassion score relative to some other group does not necessarily indicate a “lack” of compassion—or a lack in any other positive attribute. A lower relative score, as quantified through our survey, may be due to a range of factors beyond the cultural differences outlined above. For example, compassion for animals may be partly supplanted by a strong sense of compassion or empathy towards humans and their wellbeing [33], and/or a belief in the value of scientific progress.

We asked responders about their beliefs around consciousness and emotion. Unexpectedly, only 69% of responders in our survey reported that they were certain they themselves were conscious. There is no agreed definition of consciousness [35], and it is unlikely that many responders will have been taught about or reflected on the nature of consciousness. Because of this, some responders may have been unsure what capability or property we were asking them to attribute to themselves or others. Alternatively, some responders may have misread or misunderstood the question, or some may hold specific beliefs about consciousness that exclude the subjective phenomenological component. Figure 6 shows that the degree of certainty about the existence of consciousness in oneself and in other humans is stronger in UK-based responders compared to China-based responders. A qualitatively similar response is seen for rats and fruit flies too, indicating that cultural factors may be relevant here.

At any rate, the majority of responders were comfortable with attributing consciousness and emotion to humans and the possibility of consciousness and emotion in rats. This is in line with other studies where students attribute consciousness or sentience to many animals. The belief that rats and fruit flies possess emotion was positively associated with compassion scores in our survey, indicating that this belief may influence compassion, but others have shown an apparent disconnect between perceptions of sentience and the expression of compassion. For example, Paul and Podberscek [30] showed that vet students’ beliefs about animal sentience weakened for some animal species in later years of study, but this was not accompanied with a drop in self-reported empathy towards animals, at least in female students. Perhaps animals being conscious and possessing only simple emotions (rather than, for example, possessing emotions that are more complex and human-like, or possessing self-awareness, intelligence, or some other cognitive capacity) is insufficient to earn compassion? Or other factors may be more critical in determining the degree of compassion students have towards animals [42,43], for example, life experiences like companion animal ownership; other relevant beliefs or values, like being vegan or perceiving rats as pests in a non-research context; or a view of the trade-off between harm to animals and benefits to humans in scientific research that weighs benefits to humans more heavily. A systematic review of the relevant literature to identify and critically explore evidence for factors relevant to human compassion towards animals could clarify this and help us understand ways to enhance compassion towards laboratory animals.

Interestingly, people’s perceptions of the moral standing of animals seem to diminish from childhood into adulthood [44]. The authors suggest that young children have an innate tendency to include animals in their moral circle, but social influences during adolescence lead to the prioritisation of humans over animals. If this is correct, it is possible that interventions could be made to sustain or reawaken a belief in the moral standing of animals, at least for some species. This raises a further question: what could an effective intervention be? A systematic review on interventions to enhance compassion in medical education found that an arts-based storytelling approach may be effective [45], raising possibilities around the use of stories or games as a way of exploring or enhancing compassion towards lab animals [46]. Video game environments especially lend themselves to self-directed learning and the exploration of complex themes involving emotion [47,48]. They are a vehicle to not just immerse the user in these themes, but also afford agency and a space to explore different facets of being compassionate. The theories behind video game design are, broadly speaking, optimised methods of teaching users how to interact with and understand complex novel environments [49,50], and we are currently developing a digital game where players’ compassion towards lab animals can be explored [51].

Our study has a number of limitations. Our sample is relatively small, cross-sectional, and non-representative of the entire student cohort: the majority of participants were Year 1 students. Self-selection bias may have been a factor, with, for example, students who have firm perceptions about their levels of compassion being more likely to participate than those who did not. The level of experience of animal work was self-reported and not precisely defined, so these may be over- or under-reported at the individual level. We made assumptions around the responders’ perceptions when asking about acceptable levels of suffering in hypothetical research studies. Our aim was to suggest an increasing level of disease severity, but this was not made explicit in the survey and the responders may have based their responses on other factors related to the disease, for example, personal experiences with a disease. Lastly, given that at least around half of the responders do not have English as their first language, the comprehension and interpretation of all parts of the survey may be an important factor. In terms of technical limitations, as described above, we have not yet carried out a test–retest to assess the survey’s reliability over time. To facilitate analysis, we converted the Likert scale responses to numerical values. For example, in the harm–benefit analysis task, mild suffering was coded as “1” and severe suffering was coded as “3”. However, it is arguable whether a single number captures the broad range of possible procedures carried out on animals (both in reality and in participants’ minds) within that particular “band” of suffering. Similarly, it is questionable whether a procedure classified as severe is a three-fold more unpleasant experience for an animal than a typical “mild” protocol.

## 5. Conclusions

Like us, animals want to flourish by fulfilling their natural capabilities, experiencing pleasure, and avoiding mental and physical suffering [7,8]. Accordingly, a culture of care and compassion ought to be integrated into undergraduate teaching and learning in biomedical sciences. Taking a compassionate approach requires an ability to quantify compassion, but compassion is a complex psychological construct with behavioural, cognitive, and affective components that is expressed in different ways and to different degrees in different people. A tool that permits quantification will be useful in exploring the factors that contribute to a compassionate approach, but also to explore interventions related to beliefs about or behaviours towards these animals, ultimately enhancing students’ and scientists’ experiences, the quality and reliability of their scientific output, and the experiences of the animals themselves.

## Figures and Tables

**Figure 1 animals-13-03584-f001:**
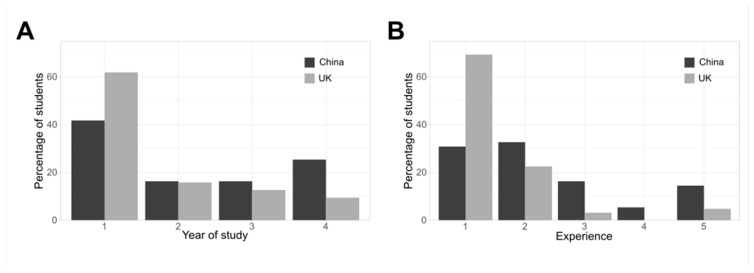
Participant demographic data. (**A**) Responders’ year of study by site: University of Edinburgh-based (UK) or ZJE-based (China). (**B**) Responders’ self-reported experience of working with laboratory animals, by site: “1” is “no experience at all”; “5” is a relatively high level of experience (see Appendix A for more detail).

**Figure 2 animals-13-03584-f002:**
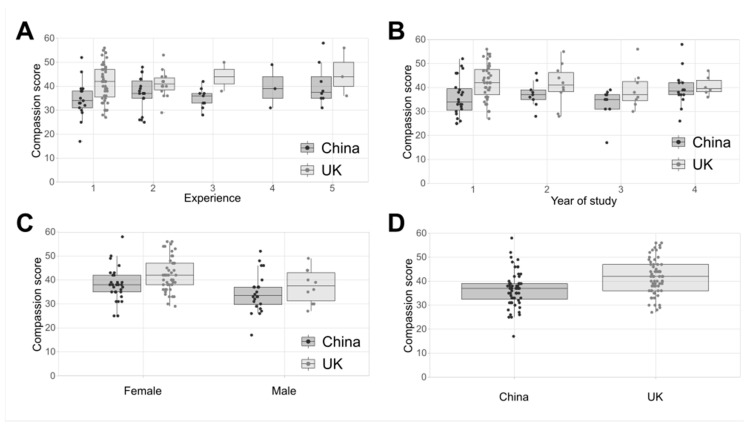
Compassion scores shown in Tukey box and whisker plots by (**A**) year of study, (**B**) gender, and (**C**) experience at both sites. (**D**) Overall compassion scores by site.

**Figure 3 animals-13-03584-f003:**
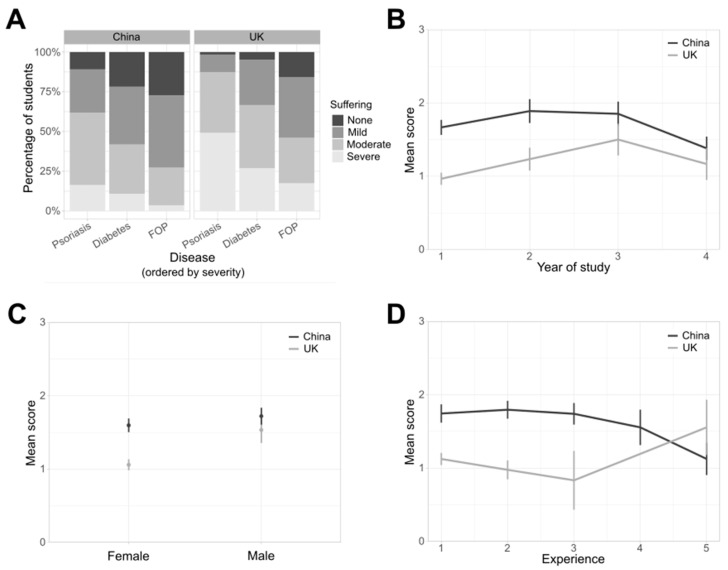
(**A**) Percentage of students at the two sites who would allow animal suffering at different severity levels in imagined research programmes using rats. The research programmes would prevent the onset of either psoriasis, type 2 diabetes, or fibrodysplasia ossificans progressiva (FOP). Mean levels of acceptable suffering by (**B**) year of study, (**C**) gender, or (**D**) experience.

**Figure 4 animals-13-03584-f004:**
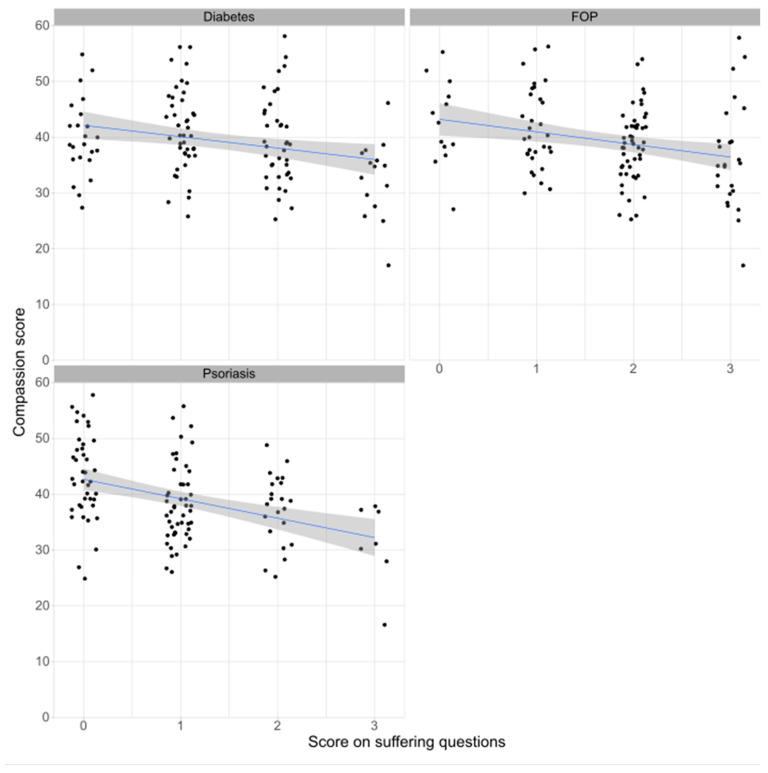
Correlations between compassion scores and acceptable animal suffering. Blue lines show linear fit, grey bands represent 95% confidence intervals.

**Figure 5 animals-13-03584-f005:**
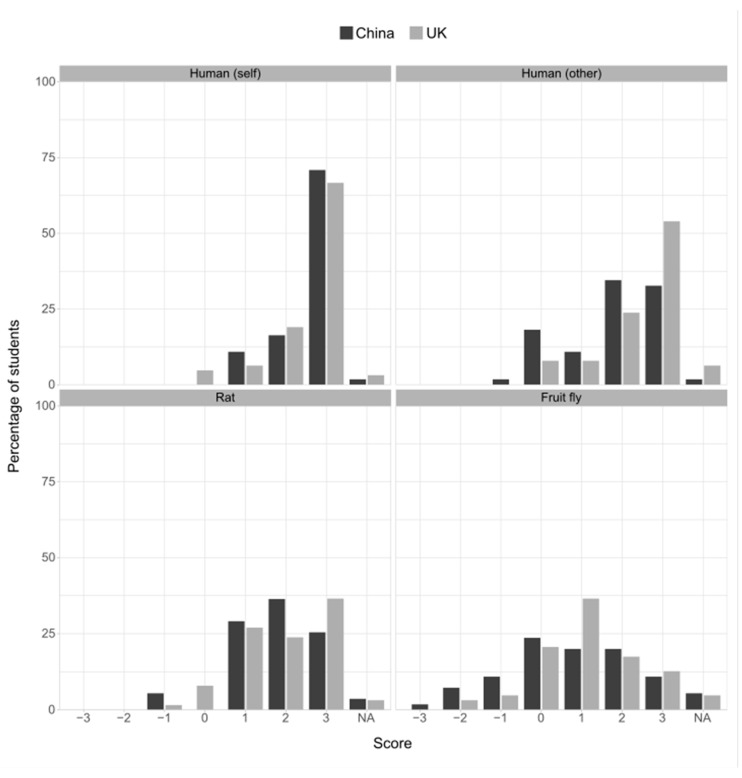
Belief in the existence of phenomenal consciousness in responders themselves (self), in other humans, in rats, or in fruit flies, expressed by site. Positive scores indicate the strength of belief, negative scores indicate the strength of non-belief. A score of 0 indicates the response “I do not know whether or not…”. “N/A” indicates the response “It is impossible for me to know whether or not…”.

**Figure 6 animals-13-03584-f006:**
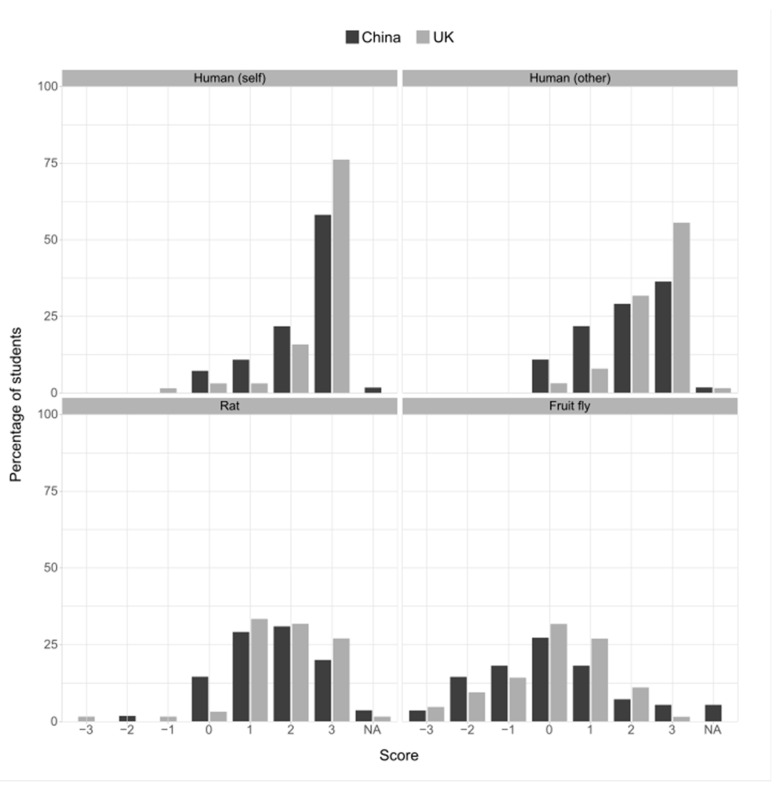
Belief in the capability to have emotion in responders themselves (self), in other humans, in rats, or in fruit flies, expressed by site. Positive scores indicate the strength of belief, negative scores indicate the strength of non-belief. A score of 0 indicates the response “I do not know whether or not…”. “N/A” indicates the response “It is impossible for me to know whether or not…”.

**Figure 7 animals-13-03584-f007:**
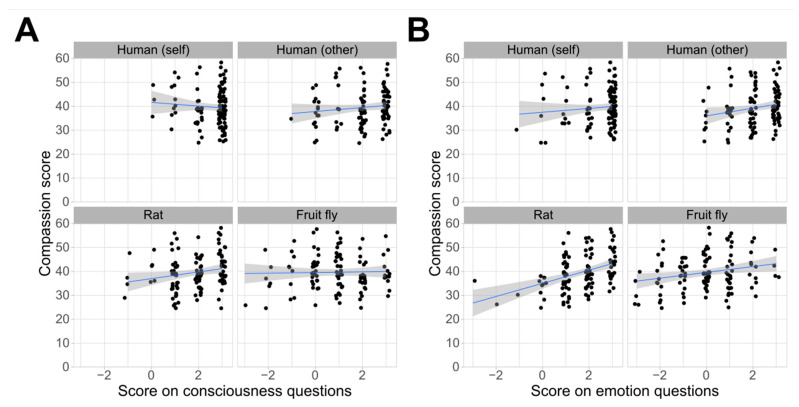
Correlations between compassion scores and beliefs about (**A**) consciousness or (**B**) the capability to have emotion in responders themselves, in other humans, in rats, and in fruit flies.

**Table 1 animals-13-03584-t001:** Association of student characteristics with compassion scores. Reference levels for categorical variables are indicated in parentheses.

Variable	Odds Ratio	CI 2.5%	CI 97.5%	*p* Value
Site (ref = UK)				
China	0.718	0.575	0.897	0.004
Gender (ref = Female)				
Male	0.768	0.613	0.962	0.023
Year of study	0.950	0.861	1.048	0.306
Experience	1.103	1	1.216	0.052

**Table 2 animals-13-03584-t002:** Association of student and disease characteristics with scores in questions related to acceptable animal suffering. Reference levels for categorical variables are indicated in parentheses.

Variable	Odds Ratio	CI 2.5%	CI 97.5%	*p* Value
Num people affected	1.09	0.81	1.47	0.567
Severity	3.20	2.28	4.49	0.000
Site (ref = UK)				
China	4.89	2.08	11.50	0.000
Year of study	1.29	0.88	1.88	0.186
Experience	0.72	0.50	1.04	0.079
Gender (ref = Female)				
Male	2.02	0.85	4.81	0.112

**Table 3 animals-13-03584-t003:** Association of species and student characteristics with scores in questions related to consciousness and emotion. Reference levels for categorical variables are indicated in parentheses.

Consciousness				
Variable	Odds Ratio	CI 2.5%	CI 97.5%	*p* Value
Species (ref = self)				
Other humans	0.183	0.098	0.341	9.13 × 10^−8^
Rat	0.098	0.052	0.184	3.83 × 10^−13^
Fruit fly	0.012	0.006	0.025	8.98 × 10^−32^
Site (ref = UK)				
China	0.456	0.179	1.160	0.099
Year	1.537	1.003	2.356	0.049
Experience	0.958	0.638	1.439	0.836
Gender (ref = Female)				
Male	1.114	0.419	2.964	0.828
**Emotion**				
**Variable**	**Odds Ratio**	**CI 2.5%**	**CI 97.5%**	***p*** **value**
Species (ref = self)				
Other humans	0.355	0.202	0.623	3.09 × 10^−4^
Rat	0.110	0.062	0.195	6.32 × 10^−14^
Fruit fly	0.006	0.003	0.013	8.41 × 10^−41^
Site (ref = UK)				
China	0.418	0.220	0.795	0.008
Year	1.425	1.059	1.917	0.019
Experience	1.072	0.806	1.424	0.634
Gender (ref = Female)				
Male	0.891	0.460	1.724	0.732

## Data Availability

The raw data for this study cannot be shared. This is a condition of the ethical approval, applied to reduce the likelihood of participant identification by linking demographic data. However, a mock data set and the code used in the analysis is available at https://github.com/nicolaromano/Fitzpatrick_2023_Compassion (accessed on 17 November 2023).

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
