# Peer review of "Exploring Compassion towards Laboratory Animals in UK- and China-Based Undergraduate Biomedical Sciences Students"

_animals, 2023, doi:10.3390/ani13223584_

Round 1
Reviewer 1 Report
Comments and Suggestions for Authors
1. Although ms. concerned attitudes regarding compassion, the idea of JUSTICE should also be considered, or at least mentioned.
2. Far from obvious that even a 'simple' utilitarianism leads to permissibility of animal research even in most circumstances. Many philosophers would disagree.
3. What is explanation that only 86% (l. 241) or 69% (l. 314) of participants think of themselves as being conscious!! To respond to this question may in and of itself require consciousness.
4. Acknowledgment that suffering may not be the only source or reason for compassion mentioned once (l. 378) where curtailment of capabilities is mentioned; more attention may be helpful here. (Nussbaum has written extensively on this).
Author Response
Comment 1. Although ms. concerned attitudes regarding compassion, the idea of JUSTICE should also be considered, or at least mentioned.
and
Comment 4. Acknowledgment that suffering may not be the only source or reason for compassion mentioned once (l. 378) where curtailment of capabilities is mentioned; more attention may be helpful here. (Nussbaum has written extensively on this).
Curtailment of capabilities is an important emerging concept in lab animal welfare, and we have added to manuscript to draw attention to it (from line 50). This relates to other interesting and relevant writings; among many: Martha Nussbaum’s critique of Rawls regarding justice for animals, Christine Korsgaard’s ideas on the relative importance of - and our obligations to - other animals, Mark Rowlands on the moral status of animals, and Eva Meijer’s theory of animal political voice. However, all of these fall outside the direct scope of this particular manuscript and while we recognise their importance, we have chosen not to elaborate on them here. On a personal note, in the context of our own (JM’s) translational research work with rats, we are very much open to these ideas, and are beginning to explore giving animals choices (rather than imposing interventions) and developing ways to explore – probably in a play format – the possibility of co-operative activity between researchers and rats. The overall aim is to explore the idea of the animal’s own motivations/intentions/agency rather than seeking to represent these as ‘models’ of human behaviour.
Comment 2. Far from obvious that even a 'simple' utilitarianism leads to permissibility of animal research even in most circumstances. Many philosophers would disagree.
We also disagree that taking a utilitarian approach necessarily permits animal use. The point we wanted to make (and it was perhaps not particularly well-made in the original manuscript) is that a human-centric utilitarian argument - namely that human welfare outweighs animal suffering, and that’s the end of the discussion - is sometimes used in biomedical teaching/training without a great deal of scrutiny, and that this is not a good thing. We have added to the manuscript to clarify this (from line 71).
Comment 3. What is explanation that only 86% (l. 241) or 69% (l. 314) of participants think of themselves as being conscious!! To respond to this question may in and of itself require consciousness.
We agree that this is puzzling. As you say, it’s hard to see how a non-conscious entity could answer a question about consciousness. We expected responders to have a range of views about other animals' consciousness, and this proved to be the case, but we expected almost all responders to report themselves as conscious. The fact that so many responders have some degree of doubt about their own consciousness is hard to explain. There may a number of reasons for this, and we outlined some in the original manuscript, but none seem particularly satisfactory. We note that Figure 6 indicates that certainty about consciousness is higher in the UK-based students compared to the China-based students for both self and for other humans, and something qualitatively similar is seen for the other species so there may be cultural factors at play too, and we added to the manuscript to highlight this observation (from line 325).
Reviewer 2 Report
Comments and Suggestions for Authors
please remove the full stop after the 86% of line 241.
A perfect publication. Thank you very much for this work
Author Response
Thank you for the positive and supportive comments. We have corrected the typo on Line 241 (line 250 in the revised version).
Reviewer 3 Report
Comments and Suggestions for Authors
In this article the authors address the essential notion of compassion for animals, and surveyed two samples of Life Science students, one based in the UK and the other based in China, using a 12-item questionnaire.
The introduction provides an appropriate scientific background. The Material and Methods section provides an extensive description of the methodology used and the choices made to successively address compassion for the animal, the level of experience with laboratory animals, the notion of harm/benefit, as well as consciousness and emotions in humans and animals.
The results are well presented with an appropriate set of figures.
Using this survey, the authors show that compassion for animals depends less on the level of study than on gender, with female students showing greater compassion than male students. In addition, British students appear to have higher compassion scores than Chinese students. These results are in line with previous studies that have demonstrated a gender effect as well as a 'cultural' effect.
Interestingly, and thanks to the robustness of the GLM model, the authors show that the level of acceptability of animal suffering was higher among Chinese students than among their British counterparts.
Finally, the authors present the results on the notion of consciousness and emotion in humans and animals. The data shows gradation in responses for humans, rats and Drosophila, with a positive correlation between compassion scores and belief in emotional capacities.
This is a truly original piece of work in that no study has previously addressed the students’ empathy or compassion towards animals. In this respect, the article deserves a great deal of attention. In my opinion, there are no major criticisms of this manuscript.
I found it particularly ambitious to survey students on the notion of harm/benefits. As someone who is familiar with ethical review of projects involving animals, I must admit that even with subjects who are older and more experienced in animal research, the harm/benefit analysis is difficult to grasp. Nevertheless, the results obtained on this question appear particularly interesting, robust, and comforting.
On both consciousness and emotions, the data also show a small percentage of respondents who say they don't know, or that it is impossible for them to know. It would be interesting to explore this aspect further and determine whether this is a properly scientific attitude, i.e admitting an absence of evidence or true ignorance.
The discussion is particularly well-constructed and relevant. The limitations of the methodology are aptly mentioned, including the language used for the two populations and the difficulty of conceptualising the harm/benefit analysis, particularly in the younger subjects, who are over-represented in the sample.
As the authors point out, it is possible to observe a reduction in compassion with time and practice, which makes the idea of proposing this questionnaire at different stages of the curriculum particularly interesting. Compassion is clearly an evolving feeling or attitude that is shaped by our experience and own emotions, with occasional incidents that upset our moral convictions or trigger defence mechanisms.
The tool presented seems to me to be extremely useful and promising. As a researcher but also a trainer in ethics and laboratory animal science, I find this questionnaire very relevant and inspiring for raising awareness of animal ethics among students and more experienced people.
While compassion for animals can fluctuate considerably over the course of a career, in one direction or another, the obvious conclusion is that we need to work to integrate awareness of animal consciousness into education as early as possible.
Author Response
We thank the reviewer for the positive comments. We respond to points raised below.
On both consciousness and emotions, the data also show a small percentage of respondents who say they don't know, or that it is impossible for them to know. It would be interesting to explore this aspect further and determine whether this is a properly scientific attitude, i.e., admitting an absence of evidence or true ignorance.
This is an interesting question that we sought to explore indirectly in the survey’s answer options. Our intention – though we are aware not all participants may have realised this – was that “it is impossible for me to know” is a positive statement reflecting the responder’s belief that it’s not currently possible by any means (e.g., by philosophical or scientific inquiry) to know what another sentient individual’s experience is like. In other words, the responder believes that others’ mental states are inaccessible. We acknowledge this is potentially a dead end in terms of further study of consciousness, but we wanted to give participants the opportunity to answer in that way. In contrast, the “I don’t know” option was for responders who did not feel able to answer the question, for any reason, with a reasonable degree of confidence. We acknowledge these are subtle (even hidden) meanings/differences and it is difficult to know what responders really meant when they selected them. Discussion with responders would be the best way to tease out what they feel/mean regarding not knowing. But because we did not do this, we did not feel it was reasonable to comment on this explicitly in the manuscript.
As the authors point out, it is possible to observe a reduction in compassion with time and practice, which makes the idea of proposing this questionnaire at different stages of the curriculum particularly interesting. Compassion is clearly an evolving feeling or attitude that is shaped by our experience and own emotions, with occasional incidents that upset our moral convictions or trigger defence mechanisms.
We agree, and as we mention in the discussion, applying this tool in other groups and/or longitudinally would be of interest, particularly if predictions can be made and tested regarding the effects on beliefs and values of specific incidents, or of cumulative experience and/or of deliberate interventions (e.g., training).
The tool presented seems to me to be extremely useful and promising. As a researcher but also a trainer in ethics and laboratory animal science, I find this questionnaire very relevant and inspiring for raising awareness of animal ethics among students and more experienced people. While compassion for animals can fluctuate considerably over the course of a career, in one direction or another, the obvious conclusion is that we need to work to integrate awareness of animal consciousness into education as early as possible.
We are very pleased to hear these supportive comments, and we hope others will use or adapt the instrument. We agree the integration of current understanding of animal capabilities (and the potential ethical impact of this) into biomedical education is important. We have begun to develop an educational resource (the digital game we refer to in the manuscript) that lets students explore lab rodent behaviour/cognition/perception and may provoke, in some players at least, reflection on how they choose to interact with laboratory animals in real life. This work is still at an early stage, but we have already had engagement and input from Learning Technologists at the Medical School and Vet School in Edinburgh, and from colleagues responsible for training and monitoring lab animal users across the University.